# Synthesis of New Proteomimetic Quinazolinone Alkaloids and Evaluation of Their Neuroprotective and Antitumor Effects

**DOI:** 10.3390/molecules24030534

**Published:** 2019-02-01

**Authors:** Solida Long, Diana I. S. P. Resende, Anake Kijjoa, Artur M. S. Silva, Ricardo Fernandes, Cristina P. R. Xavier, M. Helena Vasconcelos, Emília Sousa, Madalena M. M. Pinto

**Affiliations:** 1Laboratory of Organic and Pharmaceutical Chemistry (LQOF), Department of Chemical Sciences, Faculty of Pharmacy, University of Porto, Rua de Jorge Viterbo Ferreira, 228, 4050-313 Porto, Portugal; up201502099@ff.up.pt (S.L.); dresende@ff.up.pt (D.I.S.P.R.); madalena@ff.up.pt (M.M.M.P.); 2Interdisciplinary Centre of Marine and Environmental Research (CIIMAR), 4450-208 Matosinhos, Portugal; ankijjoa@icbas.up.pt; 3ICBAS-Instituto de Ciências Biomédicas Abel Salazar, University of Porto, 4050-313 Porto, Portugal; 4Organic Chemistry and Natural Products Unit (QOPNA), Department of Chemistry, University of Aveiro, 3810-193 Aveiro, Portugal; artur.silva@ua.pt; 5Instituto de Investigação e Inovação em Saúde (i3S), University of Porto, 4200-135 Porto, Portugal; ricardojorgefernandes@gmail.com (R.F.); cristinax@ipatimup.pt (C.P.R.X.); hvasconcelos@ipatimup.pt (M.H.V.); 6Cancer Drug Resistance Group, Institute of Molecular Pathology and Immunology of the University of Porto (IPATIMUP), 4200-135, Porto, Portugal; 7Laboratory of Microbiology, Department of Biological Sciences, Faculty of Pharmacy, University of Porto, 4050-313 Porto, Portugal

**Keywords:** antitumor, neuroprotection, quinazolinones, fiscalin B, fumiquinazoline, enantioselectivity

## Abstract

New quinazolinone derivatives of the marine-derived alkaloids fiscalin B (**3**) and fumiquinazoline G (**1**), with neuroprotective and antitumor effects, were synthesized. Eleven quinazolinone-containing indole alkaloids were synthesized, proceeding the *anti* analogs via a one-pot method, and the *syn* analogs by the Mazurkiewicz-Ganesan approach. The neuroprotection capacity of these compounds on the rotenone-damage human neuroblastoma cell SH-SY5y was evaluated using the MTT assay. Compounds **1**, **3**, **5**, and **7** showed more than 25% protection. The antitumor activity was investigated using the sulforhodamine B assay and some compounds were tested on the non-malignant MCF-12A cells. Fumiquinazoline G (**1**) was the most potent compound, with GI_50_ values lower than 20 µM. Compounds **5**, **7**, and **11** were more active in all tumor cell lines when compared to their enantiomers. Compounds **5**, **7**, **10**, and **11** had very little effect in the viability of the non-malignant cells. Differences between enantiomeric pairs were also noted as being essential for these activities the *S*-configuration at C-4. These results reinforce the previously described activities of the fiscalin B (**3**) as substance P inhibitor and fumiquinazoline G (**1**) as antitumor agent showing potential as lead compounds for the development of drugs for treatment of neurodegenerative disorders and cancer, respectively.

## 1. Introduction

The pathophysiology of neurodegenerative diseases is poorly understood, and there are few therapeutic options, making neuroprotective drug discovery appealing for medicinal chemists. Although cancer and neurodegeneration have very distinct pathological disorders, over recent years growing evidence indicates that they share common molecular pathways [1]. Furthermore, it is recognized that several drugs used in the treatment of neurodegenerative diseases display antitumor effects while some antitumor drugs are neuroprotective [2].

Marine-derived indolylmethylpyrazinoquinazoline alkaloids, with a pyrazino[2,1-*b*]quinazoline-3,6-dione linked to an indole moiety (Figure 1), have attracted our attention due to their promising antitumor activities [3], with *epi*-fiscalin A [4], fumiquinazoline A [5,6,7,8,9], fumiquinazoline G [7], and versiquinazolines [10] as the most active analogs. Moreover, the response of fiscalins A-C [11] and (−)-spiroquinazoline [12] (Figure 1A) as substance P inhibitors was also reported as a novel neuroprotective therapy in the intrastriatal 6-hydroxydopamine model of early stage of Parkinson’s disease (PD) [13]. It is well known that among compounds implicated in neurodegeneration, non-proteinogenic amino acids may cause significant collateral neurodegenerative damage [14]. Rodgers *et* al. [15] reported that proteomimetic l-tyrosine of l-DOPA is cytotoxic in vitro and capable of generating protein aggregation, whereas non-protein amino acid β-methylamino-l-alanine (BMAA) has been linked to neurological diseases such as amyotrophic lateral sclerosis (ALS) and PD since BMAA was detected in brain protein of LAS and PD patients. In the previous work, we have described syntheses of a series of fiscalin B derivatives, which showed weak to moderate antitumor activity against non-small cell lung cancer (NCI-H460) and colorectal adenocarcinoma (HCT-15) cell lines [16]. These findings led us to develop a small library of proteomimetic quinazolinone-derived compounds (Figure 1B) with different configurations at C-1 and C-4 to investigate their action on neurodegenerative disorders as well as to further explore their potential as tumor cell growth inhibitors, putting in evidence the influence of the stereochemistry of the derivatives.

## 2. Results

### 2.1. Chemistry

Two synthetic approaches were used to prepare the *syn* and *anti* enantiomers of quinazolinone alkaloids. The *syn* enantiomers **1** (fumiquinazoline G) and **2** were synthesized by the Mazurkiewicz-Ganesan procedure [17] (Scheme 1A) by coupling anthranilic acid (**i**) with D-tryptophan methyl ester (**ii**) for **1** or with l-tryptophan methyl ester (**vi**) for **2,** using 1,1,3,3-tetramethylaminium tetrafluoroborate (TBTU) in alkaline condition to obtain the dipeptide **iii** or **vii**. Then, the coupling of **iii** or **vii** with N-protected α-amino acid chloride in a two-phase Schotten-Baumann condition yielded a tripeptide (dehydrate β-keto amides) **v** or **ix**. The oxazole intermediates were obtained by adding the dehydrating agent, triphenylphosphine (Ph_3_P), and I_2_ to dehydrate β-keto amide **v** or **ix**, and N-deprotection by 20% piperidine afforded **1** and **2**. On the other hand, a highly effective and environmentally friendly approach using a microwave-assisted multicomponent polycondensation of amino acids was used to prepare a series of the *anti* enantiomers of pyrazinoquinazoline alkaloids [18], as described in our previous work [16]. This methodology was used to synthesize new derivatives of fiscalin B (**3**) and fumiquinazoline G (**1**), **4, 5, 6**, **7**, and **8** (Scheme 1B). The *syn* isomer **9** was obtained along with **8,** and both were isolated by preparative thin layer chromatography (TLC). Diastereoisomers of **10** and **11** were obtained after deprotection of O-benzyl group from **8** and **9**, respectively, using boron trichloride, according to Okaya et al. [19] with a slight modification. Compound **12** was also synthesized using microwave irradiation from 3,5-dichloroanthranilic acid (**xiii**). The purity of the compounds was determined by a reversed-phase liquid chromatography (LC, C18, MeOH:H_2_O; 60:40 or CH_3_CN:H_2_O; 50:50) and was found to be higher than 90%. The enantiomeric ratio (er) was determined by a chiral LC equipped with amylose tris-3,5-dimethylphenylcarbamate column, using hexane:EtOH (80:20) or (70:30) as a mobile phase. 

The reaction carried out using microwave with high temperature resulted not only in low yields of the products in the range of 2.2 to 21.7%, but also with a high degree of epimerization (Scheme 1). Contrary to what has been found in our previous study [16] that the reaction under a microwave irradiation was regioselective and yielded only *anti* isomers, the synthesis of **8**, by a microwave irradiation, produced also its *syn* epimer, **9** [4-(benzyloxy)-1-methylbenzyl at C-1], with a 22% yield. This study suggested that microwave irradiation is beneficial for the synthesis of quinazolinone alkaloids with bulky substituents at C-1 which was previously reported as unsuccessful by Mazurkiewicz-Ganesan method [17]. However, this methodology failed for the synthesis of *syn* enantiomers as described in the experimental section for **4** and **6**. The *syn* enantiomers of **1** and **2** were synthesized by Mazurkiewicz-Ganesan approach [17] and gave moderate yields (37 and 26%, respectively). Compounds **10** and **11** were obtained by deprotection in good yields (30 and 69%, respectively).

Moreover, the methodology involving microwave irradiation was characterized by producing partial epimerizations. Surprisingly, **4** and **5**, with three stereogenic centers, gave a higher enantiomeric ratio (er) of 99%. Similar to the previous report for fiscalin B analogs [16], the multi-step approach gave a better yield and, in most cases, higher enantiomeric ratios due to milder conditions; nonetheless, the one-pot reaction is a faster alternative to provide *anti* enantiomers with diversity of substituents at C-1.

### 2.2. Structure Elucidation

A series of 1D and 2D NMR experiments and HRMS were used to confirm the structures of all the new compounds (Appendix A). The amide proton (H-2) appeared as a broad singlet (*brs*) in the *anti* isomers, i.e., **4**, **5**, **6**, **7**, **8**, **10**, and **12,** but as a doublet (*d*) in the *syn* isomers, i.e., **1**, **2**, **9**, and **11** (Appendix A). Each stereoisomer exhibited different chemical shift values for H-2, H-4, and H-1. For example, for isomers **2**, **4**, and **5,** with the *iso*butyl group at C-1, the *syn* and the *anti* isomers were distinguished by the chemical shift values of H-1 and H-1’; which were *ca.* δ_H_ 4.03 and 0.93, respectively for the *syn* isomer (**2**), and at *ca.* δ_H_ 2.80 and 2.3, respectively, for the *anti* isomers (**4** and **5**), due to the absence of the shielding effect by the aromatic ring of the indole moiety [20]. 

This assignment was also confirmed by HMBC correlations which distinguished between the isomers of **2**, **4**, and **5** by the presence or absence of correlations from H-2 to C-3, C-4, and C-14. In **2,** H-2 showed no correlation to C-3, C-4, and C-14 whereas in **4** and **5,** H-2 showed correlations to those three carbons. H-4 also showed correlations to different carbons among isomers. H-4 showed correlations to C-3 and C-4’ in **2**, but to C-3, C-14, C-4’, and C-5’ in **4**, while there were no such correlations observed in **5**. H-1 of the *syn* isomer of **2** showed correlation only to C-14 while in the *anti* isomer of **4** and **5**, it displayed correlations to C-14, C-1’ and C-3” (Figure 2)**.** In contrast, in **1**, H-2 and H-4 showed no correlations to carbons that are two or three bonds away (^2^*J* or ^3^*J*), and H-1 showed cross peaks to C-3 and C-14. Furthermore, H-8 and H-10 of **12** (with Cl at C-9 and C-11) appeared as two doublets (δ_H_ 8.26, *J* = 2.4 Hz and δ_H_ 7.85, *J* = 2.4 Hz, respectively) while in other compounds H-8 appeared as a double doublet and H-10 as double-double doublet. The NOESY spectrum for compound **6** with an *anti* configuration (*1R*, *4S*) showed correlations from H-1 to H-1’ and H-3’ (methyl group attached to S atom) and H-2, while H-4 showed cross peak to H-4’ (Appendix A). The NOESY spectrum for the *syn* configuration compound **2** (*1S*, *4S*) exhibited correlations between H-1 and H-1’, H-2’, H-3’’, and H-2 as well as from H-4 to H-4’. Also, correlations between H-4’ and H-1’ in the *syn* configuration compound **2** could be noted while these was absent in the *anti* configuration compound **6** (Appendix A). These observations were similar to the previously described for fiscalin B isomers [16].

### 2.3. Neuroprotection Activity

The neuroprotection assay was performed on human neuroblastoma cell SH-SY5y treated with rotenone, a toxin that acts by interfering with the electron transport chain in mitochondria, inhibiting the transfer of electron from iron-sulfur centers in complex I to ubiquinone. This in turn interferes with NADH, therefore creating reactive oxygen species (ROS), which can damage DNA and other components, leading to cell death [21,22]. In animal experimentation, rotenone reproduces features of PD, including selective nigrostriatal dopaminergic degeneration and alpha-synuclein-positive cytoplasmic inclusions [23]. Furthermore, rotenone triggers mitochondrial impairment, oxidative damage, and cell death in neuronal culture, phenomena that are common in neurodegenerative diseases [24].

In this assay, the SH-SY5y cells were treated with 2 µM of rotenone for 24 h. The MTT assay, which assesses cell metabolic activity through the activity of NAD(P)H-dependent cellular oxidoreductase enzyme that reflect the number of viable cells [24], was used for quantifying the cell death. The cellular protection of **1**, **2**, **4**, **5**, **6**, **7**, **8**, **9**, **10**, **11**, and **12** against the toxin was determined by MTT assay and expressed as percentage referred to the cell treated with rotenone at 10 different concentrations to produce the dose-dependent curve. Synthetic fiscalin B (**3**), previously obtained by some of us [16] and was reported as a substance P antagonist [12], was also tested in this assay for comparison. Compounds considered as neuroprotective must have (i) more than 25% of protection, (ii) statistically significant difference, and (iii) protection is more than one dose [23,24].

Most of the compounds, at the highest concentration tested (100 µM), were found to increase toxicity, which ensures that the compounds were assayed at their maximal tolerated dose (MTD). Compounds **1**, **2**, **3**, **4**, **5**, **10**, **11**, and **12** showed high toxicity at 100 µM and some of them also at 50 µM. The compounds which showed neuroprotective activity in rotenone-treated in vitro model were **1**, **3**, **5**, and **7**. Additionally, some neuroprotective effect was also observed for **11** while **2**, **4**, **6**, **8**, **9, 10**, and **12** did not display any neuroprotection (Figure 3). Compound **1**, **3**, **5**, and **7** showed more than 25% of protection of the cell death at least in one concentration. Compound **7** exhibited the best neuroprotective activity, with rotenone inhibitory of 47, 40, 42, 39, and 31% at the concentrations of 1.56, 3.13, 6.25, 12.5, and 25 µM, respectively.

### 2.4. Tumor Cell Growth Inhibitory Activity

Compounds **1**–**2**, **4**–**8**, **10** and **11** were tested for tumor cell growth inhibitory activity on three human tumor cell lines: NCI-H460 (non-small cell lung cancer), BxPC3 (human pancreatic adenocarcinoma), and PANC1 (human pancreatic adenocarcinoma) using the sulforhodamine B (SRB) colorimetric assay [16,25]. Cells were exposed to five concentrations of each compound (at a maximum concentration of 25, 150 or 200 μM, depending on the compound) for 48 h. Doxorubicin was used as a positive control for NCI-H460 cell line, and gemcitabine was used as a positive control for the BxPC3 and PANC1 cell lines. The antitumor activity was reported as GI_50_ concentration (drug concentration that inhibits the growth of cancer cells by 50%).

Compounds **2**, **4**–**8, 10**, and **11** showed weak to moderate growth inhibitory, with GI_50_ ranking from 27.93 ± 0.8 to 151.07 ± 2.9 µM (Table 1). In general, compounds with the indolylmethyl substituent on C-4 whose configuration of C-4 is *R* showed better antitumor activity in all cell lines when compared to those with 4*S* configuration. This was evidenced by stronger antitumor activities of **1**, **5**, and **7** than those of **2**, **4**, and **6**. Only fumiquinazoline G (**1**) with *R*-configuration for both C-1 and C-4 showed strong growth inhibitory effect in all cancer cell lines tested (GI_50_ ranging from 7.62 ± 0.7 to 17.34 ± 1.7 µM). In addition, these results agree with those reported in our previous publication in which enantiomers with *R*-configuration at C-1 and C-4 showed better antitumor activity than enantiomers with *S*-configuration [16]. Unfortunately, **9** and **12** tested in this study could not be evaluated regarding their tumor cell growth inhibitory activity due to contamination of the compounds (data not shown).

### 2.5. Activity in Non-Tumor Cells

Compounds presenting the best neuroprotection and/or antitumor effects, namely **5**, **7**, **10**, and **11,** were also evaluated against the non-malignant MCF-12A human breast epithelial cells. For that, one concentration of each compound (corresponding to approximately the highest GI_50_ value obtained in the cell growth inhibitory activity assay) was tested and the percentage of cell growth inhibition was determined by the SRB assay. In this assay, the duration of the SRB assay had to be longer since non-malignant cells have a much slower growth rate than tumor cells. This longer duration of the assay also allowed to evaluate possible delayed effects of the compounds in these non-malignant cells. Therefore, the assay in the MCF-12A cells was performed following 7 days of treatment with compounds (48 h with compound incubation plus 5 days without the compounds). As shown in Table 2, all the tested compounds caused a small effect in the growth of these non-malignant cells, meaning that the cell growth inhibition detected in MCF-12A cells after 7 days of treatment was much lower than that detected in the tumor cell lines after 2 days of treatment (when tested at the same concentration).

## 3. Structural-Activity Relationship (SAR)

Structure-activity relationship analysis showed that the obtained results were consistent with data previously reported for the natural product fumiquinazoline G [26] and fiscalin B [11,27,28,29] and their derivatives [16] (Figure 4). Moreover, it was found that the configurations of C-1 and C-4 have strong influence on antitumor activity since fumiquinazoline G (**1**), with *R-*configurations at C-1 and C-4, showed the strongest antitumor effect against NCI-H460, BxPC3, and PANC1 cell lines. Comparing the enantiomeric pairs **4** and **5,** compound **5** with also *R-*configuration at C-4 and *S* at C-1 showed stronger antitumor effect. In contrast, their isomer, compound **2**, with *S-*configurations both at C-1 and C-4 were *S* exhibited the weakest inhibitory effect on all cell lines. In addition, the substituent at C-1 also affected the antitumor activity. Alkyl residues (isoleucine residues) in **2**, **4**, and **5** showed better antitumor activity than that found in compounds with a sulfur atom (methionine residues) such as **6** and **7**. Aromatic groups such as the tyrosine residue present in **10** and **11** produced good antitumor activity, with GI_50_ values ranging from 27.93 ± 0.8 to 50.59 ± 2.3 µM, but the substitution by benzyl groups as in the case of **8** caused a 2-fold decrease in the antitumor activity. Regarding neuroprotection capacity, SAR suggests that the *R-*configuration at C-4 is also important (when comparing compounds **6** and **7**, Figure 3); however, increasing the molecular weight of C-1 substituent has a negative effect in neuroprotection. In addition, this study confirmed that quinazolinone alkaloids which act as substance P inhibitors (i.e., fiscalin B, **3**), showed a potential as neuroprotective agents. Therefore, the studied fumiquinazoline-derived alkaloids showed promising antitumor and neuroprotection effects and deserves to be further explored.

## 4. Materials and Methods 

### 4.1. General Procedure

All reagents were from analytical grade. Dried pyridine and triphenylphosphite were purchased from Sigma (Sigma-Aldrich Co. Ltd., Gillingham, UK). Anthranilic acid (**i**) and Protected amino acids (**ii** and **vii**) were purchased from TCI (Tokyo Chemical Industry Co. Ltd., Chuo-ku, Tokyo, Japan). Column chromatography purifications were performed using flash silica Merck 60, 230–400 mesh (EMD Millipore corporation, Billerica, MA, USA) and preparative TLC was carried out on precoated plates Merck Kieselgel 60 F_254_ (EMD Millipore corporation, Billerica, MA, USA), spots were visualized with UV light (Vilber Lourmat, Marne-la-Vallée, France). Melting points were measured in a Köfler microscope and are uncorrected. Infrared spectra were recorded in a KBr microplate in a FTIR spectrometer Nicolet iS10 from Thermo Scientific (Waltham, MA, USA) with Smart OMNI-Transmission accessory (Software 188 OMNIC 8.3, Thermo Fisher Scientific Inc., Austin, TX, USA). ^1^H and ^13^C NMR spectra were recorded in CDCl_3_ or DMSO-d_6_ (Deutero GmbH, Kastellaun, Germany) at room temperature unless otherwise mentioned on Bruker AMC instrument (Bruker Biosciences Corporation, Billerica, MA, USA), operating at 300 MHz for ^1^H and 75 MHz for ^13^C). Carbons were assigned according to HSQC and or HMBC experiments. Optical rotation was measured at 25 °C using the ADP 410 polarimeter (Bellingham + Stanley Ltd., Royal Tunbridge Wells, Kent, UK), using the emission wavelength of sodium lamp, concentrations are given in g/100 mL. Qualitative GC-MS analyses were performed on a Trace GC 2000 Series ThermoQuest gas chromatography (Thermo Fisher Scientific Inc., Austin, TX, USA) equipped with ion-trap GCQ Plus ThermoQuest Finnigan mass detector (Thermo Fisher Scientific Inc.). Chromatographic separation was achieved using a capillary column (30 m × 0.25 mm × 0.25 μm, cross-linked 5% diphenyl and 95% dimethyl polysiloxane) from Thermo Scientific^TM^ (Thermo Fisher Scientific Inc.) and high-purity helium C-60 as carrier gas. High resolution mass spectra (HRMS) were measured on a Bruker FTMS APEX III mass spectrometer (Bruker Corporation, Billerica, MA, USA) recorded as ESI (Electrospray) made in Centro de Apoio Cientifico e Tecnolόxico á Investigation (CACTI, University of Vigo, Pontevendra, Spain). The purity of synthesized compounds was determined by reversed-phase LC with diode array detector (DAD) using C18 column (Kimetex^®^, 2.6 EV0 C18 100 Å, 150 × 4.6 mm), and the mobile phase was methanol:water (60:40) or acetonitrile:water (50:50). Enantiomeric ratio was determined by chiral LC (LCMS-2010EV, Shimadzu, Lisbon, Portugal), employing a system equipped with a chiral column (Lux^®^ 5 µm Amylose-1, 250 × 4.6 mm) and UV-detection at 254 nm, mobile phase was hexane:ethanol (80:20) and the flow rate was 0.5 mL/min. Compound **3** was obtained according to previous described method [17]. Neuroprotection studies were performed in Fundación Centro de Excelencia en Investigación de Medicamentos Innovadores en Andalucía, MEDINA (Granada, Spain).

### 4.2. General Conditions for the Synthesis of Compound (1R,4R)-4-((1H-indol-3-yl)methyl)-1-((R)-methyl)-1,2-dihydro-6H-pyrazino[2,1-b]quinazoline-3,6(4H)-dione (**1**)

To a mixture of anthranilic acid (**i**, 287 mg, 2.39 mmol) and TBTU (920 mg, 2.86 mmol, 1.2 equiv) in acetonitrile (20 mL) was added Et_3_N (833 µL, 4.78 mmol, 2 equiv) and d-tryptophan methyl ester (**ii**, 521 mg, 2.39 mmol) at room temperature. After stirring for 5 h, the reaction mixture was concentrated under reduced pressure. The residue was dissolved in CH_2_Cl_2_ and washed with 1 M HCl, extracted with CH_2_Cl_2_ (3 × 100 mL), dried with Na_2_SO_4_, filtered, and concentrated. The residue was purified by flash chromatography (eluent 1% MeOH in CH_2_Cl_2_) to yield **iii** as a white solid. ^1^H NMR and ^13^C NMR referred to the previous work [16]. To a solution of **iii** (140 mg, 0.416 mmol) in dried CH_2_Cl_2_ (10 mL) *N-*Fmoc-d-alanine-Cl [30] (**iv**,182 mg, 0.5 mmol) was added. The mixture was stirred for 30 min, followed by the addition of aqueous Na_2_CO_3_ (1 M, 8 mL, 8 mmol). After continuous stirring for 3 h, the mixture was extracted with CH_2_Cl_2_ (4 × 100 mL), dried with Na_2_SO_4_, filtered, and concentrated. The residue was purified by flash chromatography (eluent: 5% MeOH in CH_2_Cl_2_) to give **v** (220.4 mg, 84.2%) as a white solid. ^1^H NMR (300 MHz, CDCl_3_) *δ* 11.48 (s, 1H), 8.58 (d, 1H, *J* = 8.4 Hz), 8.13 (s, 1H), 7.76 (d, 2H, *J* = 7.5 Hz), 7.66 (d, 1H, *J* = 7.1 Hz), 7.59 (t, 1H, *J* = 7.4 Hz), 7.49-7.26 (m, 8H), 7.17 (t, 1H, *J* = 7.2 Hz), 7.06 (t, 1H, *J* = 7.6 Hz), 7.00 ( t, 1H, *J* = 7.7 Hz), 6.97 (s, 1H), 6.71 (d, 1H, *J* = 7.6 Hz), 5.55 (d, 1H, *J* = 6.8), 5.03 (dt, 1H, *J* = 7.6, 5.3 Hz), 4.44 (m, 2H), 4.36 (1m, 1H), 4.26 (t, 1H, *J* = 7.0 Hz), 3.73 (s, 3H), 3.40 (dd, 1H, *J* = 15.3, 5.8 Hz), 3.34 (dd, 1H, *J* = 15.3, 5.3 Hz), 1.53 (d, 3H, *J* = 7.0 Hz) and ^13^C NMR (75 MHz, CDCl_3_) 172.7, 172.2, 168.8, 156.6, 144.2, 143.8, 141.4, 138.6, 136.4, 132.8, 127.8, 127.4, 127.2, 125.3, 123.4, 123.3, 122.1, 121.6, 120.9, 120.0, 119.5, 118.3, 111.7, 109.2, 67.3, 53.6, 52.7, 52.2, 47.3, 27.3, 18.4 (See in [31]). To a solution of **v** (183.2 mg, 0.278 mmol) in dried CH_2_Cl_2_ (20 mL) Ph_3_P (365 mg, 1.39 mmol, 5 equiv), I_2_ (345 mg, 1.36 mmol. 4.9 equiv), and *N*,*N*-diisopropylethylamine (489 µL, 2.81 mmol, 10 equiv) were added. The reaction mixture was stirred at room temperature for 5 h, quenched with aqueous Na_2_CO_3_, and extracted with CH_2_Cl_2_ (3 × 100 mL), dried with Na_2_SO_4_, filtered, and concentrated. Hexane was added to remove an excess of Ph_3_P, the precipitate was filtered and was treated with CH_2_Cl_2_ (10 mL) and piperidine (2.5 mL, 20%) at room temperature for 20 min, followed by solvent evaporation to provide the solid which was triturated with hexane (1 × 200 mL), CH_2_Cl_2_/PhMe (1 × 200 mL), and hexane (1 × 200 mL). The vacuum-dried crude residue was dissolved in CH_3_CN (10 mL) in the presence of DMAP (64 mg, 0.53 mmol) and refluxed for 19 h. The reaction mixture was purified by preparative TLC (EtOAc:MeOH:CH_2_Cl_2_, 50:2.5:47.5) to afford **1**. Yield: 22.4 mg, 21.11%; *er* = 7:93; mp: 105.9–106.5 °C; ${\left[\alpha \right]}_{D}^{30}$ = −117.64 (*c* 0.034; CHCl_3_); *v_max_* (KBr) 3406, 2924, 2852, 1678, 1473, 1329, and 1261 cm^-1^; ^1^H NMR (300 MHz, CDCl_3_): *δ* 8. 38 (dd, 1H, *J* = 8.0 and 1.1 Hz, CH), 8.18 (br, 1H, NH-Trp), 7.78 (ddd, 1H, *J* = 8.5, 7.1, and 1.6 Hz, CH), 7.57 (d, 1H, *J* 7.4 Hz, CH), 7.54 (dd, 1H, *J* = 8.0 and 1.0 Hz, CH), 7.29 (d, 1H, *J* = 3.1 Hz, CH-Trp), 7.27 (d, 1H, *J* = 2.0 Hz, CH-Trp), 7.08 (ddd, 1H, *J* = 9.5, 7.0 and 0.9 Hz, CH-Trp), 6.83 (ddd, 1H, *J* = 8.7, 7.1 and 1.0 Hz, CH-Trp), 6.73 (d, 1H, *J* = 2.3 Hz, CH-Trp), 6.70 (d, 1H, *J* = 2.0 Hz, NH-amide), 5.54 (dd, 1H *J* = 5.2 and 3.6 Hz, CH*-Trp), 4.46 (qd, 1H, *J* = 6.9 and 2.8 Hz, CH*-ala), 3.78 (dd, 1H, *J* = 14.9 and 5.3 Hz, CH_2_-Trp), 3.70 (dd, 1H, *J* = 14.9 and 3.4 Hz, CH_2_-Trp), 0.58 (d, 3H, *J* = 7.0 Hz, CH_3_-ala); ^13^C NMR (75 MHz, CDCl_3_): *δ* 167.3 (C=O), 161.0 (C=O), 151.2 (C=N), 147.2 (C), 135.7 (C-Trp), 134.9 (CH), 127.9 (C-Trp), 126.9 (CH), 126.8 (CH), 126.7 (CH), 123.7 (CH-Trp), 122.3 (CH-Trp), 120.1 (C), 119.9 (CH-Trp), 118.7 (CH-Trp), 111.1 (CH-Trp), 109.4 (C-Trp), 56.9 (CH*-Trp), 51.9 (CH*-ala), 27.1 (CH_2_-Trp), 22.8 (CH_3_-ala. (+)-HRMS-ESI *m/z* 359.1505 (M + H)^+^, (calculated for C_21_H_18_N_4_O_2_, 358.1430).

### 4.3. General Conditions for the Synthesis of Compound (1S,4S)-4-((1H-indol-3-yl)methyl)-1-((S)-sec-butyl) -1,2-dihydro-6H-pyrazino[2,1-b]quinazoline-3,6(4H)-dione (**2**) 

To a mixture of anthranilic acid (**i**, 287 mg, 2.39 mmol) and TBTU (920 mg, 2.86 mmol, 1.2 equiv) in acetonitrile (20 mL), Et_3_N (833 µL, 4.78 mmol, 2 equiv) and l-tryptophan methyl ester (**vi**, 521 mg, 2.39 mmol) were added at room temperature. After stirring for 5 h, the reaction mixture was concentrated under reduced pressure. The residue was dissolved in CH_2_Cl_2_ and washed with 1 M HCl, extracted with CH_2_Cl_2_ (3 × 100 mL), dried with Na_2_SO_4_, filtered, and concentrated. The residue was purified by flash chromatography (eluent 1% MeOH in CH_2_Cl_2_) to yield v**ii** as a white solid. ^1^H NMR and ^13^C NMR referred to the previous work [16]. To a solution of v**ii** (304 mg, 0.901 mmol) in dried CH_2_Cl_2_ (30 mL), *N-*Fmoc-l-isoleucine-Cl [30] (**viii**, 395 mg, 1.08 mmol) was added. The mixture was stirred for 30 min, followed by addition of aqueous Na_2_CO_3_ (1 M, 16 mL, 16 mmol). After continuous stirring for 3 h, the mixture was extracted with CH_2_Cl_2_ (4 × 100 mL), dried with Na_2_SO_4_, filtered, and concentrated. The residue was purified by flash chromatography (eluent: 5% MeOH in CH_2_Cl_2_ to give **ix** (475.8 mg, 86.2%) as a white solid. ^1^H NMR (300 MHz, CDCl_3_): *δ* 11.43 (s, 1H), 8.59 (d, 1H, *J* = 8.3 Hz), 8.19 (s, 1H), 7.76 (d, 2H *J* = 7.4 Hz), 7.69-7.56 (m, 2H), 7.53-7.28 (m, 7H), 7.17 (t, 1H, *J* = 7.3 Hz), 7.06 (t, 1H, *J* = 7.4 Hz), 7.01 (d, 1H, *J* = 7.6 Hz), 6.96 (d, 1H, *J* = 3.5 Hz), 6.72 (d, 1H, *J* = 7.6 Hz), 5.55 (d, 1H, *J* = 8.4 Hz), 5.06 (dd, 1H, *J* = 12.6 and 5.2 Hz), 4.39 (dd, 2H, *J* = 16.4 and 9.1 Hz), 3.73 (s, 3H), 3.37 (m, 2H), 2.11-2.00 (m, 1H), 1.68-1.48 (m, 2H), 1.03 (d, 3H, *J* = 6.8 Hz), 0.96 (t, 3H, *J* = 7.3 Hz) and ^13^C NMR (75 MHz, CDCl_3_) 172.1, 170.2, 168.6, 144.1, 143.8, 141.3, 139.0, 136.1, 132.9, 127.7, 127.5, 127.1, 127.1, 126.9, 125.3, 125.2, 123.2, 122.8, 122.4, 121.4, 120.2, 120.0, 119.8, 118.5, 111.4, 109.7, 67.2, 61.3, 53.3, 52.6, 47.3, 37.9, 31.4, 27.3, 15.8, 11.7. To a solution of **ix** (291.5 mg, 0.432 mmol) in dried CH_2_Cl_2_ (20 mL) Ph_3_P (565 mg, 2.16 mmol, 5 equiv), I_2_ (448 mg, 2.12 mmol. 4.9 equiv), and *N*,*N*-diisopropylethylamine (753 µL, 4.32 mmol, 10 equiv) were added. The reaction mixture was stirred at room temperature for 5 h, quenched with aqueous Na_2_CO_3_, and extracted with CH_2_Cl_2_ (3 × 100 mL), dried with Na_2_SO_4_, filtered, and concentrated. Hexane was added to remove an excess of Ph_3_P, the precipitate was filtered and was treated with CH_2_Cl_2_ (10 mL) and piperidine (2.5 mL, 20%) at room temperature for 20 min, followed by solvent evaporation to provide the solid which was triturated with hexane (1 × 200 mL), CH_2_Cl_2_/PhMe (1 × 200 mL), and hexane (1 × 200 mL). The vacuum-dried crude residue was dissolved in CH_3_CN (10 mL) in the presence of DMAP (158 mg, 1.39 mmol) and refluxed for 19 h. The reaction mixture was purified by preparative TLC (EtOAc: MeOH: CH_2_Cl_2_, 50:2.5:47.5) to afford **2**. Yield: 36 mg, 36.2%; enantiomeric ratio (er) = 73:27; m.p: 181–183 °C; ${\left[\alpha \right]}_{D}^{30}$ = +346.40 (*c* 0.051; CHCl_3_); *v_max_* (KBr) 3375, 3187, 2880, 1684, 1662, 1472, 1434, and 1261 1 cm^−1^; ^1^H NMR (300 MHz, CDCl_3_): *δ* 8.38 (dd, 1H, *J* = 8.0 and 1.1 Hz, CH), 8.07 (br, 1H, NH-Trp), 7.79 (ddd, 1H, *J* = 8.5, 7.1, and 1.6 Hz, CH), 7.62 (dd, *J =* 8.2 and 0.5 Hz, CH), 7.54 (ddd, 1H, *J* = 8.2, 7.2, and 1.2 Hz, CH), 7.48 (d, 1H, *J* 7.9 Hz, CH-Trp), 7.28 (d, 1H, *J* = 8.5 Hz, CH-Trp), 7.12 (ddd, 1H, *J* = 8.1, 7.1 and 1.1 Hz, CH-Trp), 6.94 (ddd, 1H, *J* = 8.0, 7.1 and 1.0 Hz, CH-Trp), 6.87 (d, 1H, *J* = 2.3 Hz, CH-Trp), 6.55 (d, 1H, *J* = 3.1 Hz, NH-amide), 5.52 (dd, 1H, *J* = 6.4 and 3.6 Hz, CH*-Trp), 4.03 (dd, 1H, *J* = 8.0 and 3.5 Hz, CH-Ile), 3.83 (dd, 1H, *J* = 14.9 and 6.4 Hz, CH_2_-Trp), 3.73 (dd, 1H, *J* = 14.8 and 3.5 Hz, CH_2_-Trp), 0.99–0.85 (m, 1H, CH*-Ile), 0.85–0.69 (m, 2H, CH_2_-Ile), 0.66 (d, 3H, *J* = 6.5 Hz, CH_3_-Ile), 0.58 (t, 3H *J* = 7.1 Hz, CH_3_-Ile); ^13^C NMR (75 MHz, CDCl_3_): *δ* 167.8 (C=O), 161.4 (C=O), 149.4 (C=N), 146.8 (C), 135.9 (C-Trp), 134.7 (CH), 127.9 (C-Trp), 127.1 (CH), 127.1 (CH), 126.8 (CH), 123.5 (CH-Trp), 122.3 (CH-Trp), 120.3 (C), 120.0 (CH-Trp), 119.0 (CH-Trp), 111.0 (CH-Trp), 110.2 (C-Trp), 60.8 (CH*-Ile), 57.6 (CH*-Trp), 40.8 (CH*-Ile), 27.1 (CH_2_-Trp), 24.3 (CH_2_-Ile), 15.3 (CH_3_-Ile), 10.4 (CH_3_-Ile); (+)-HREM-ESI *m/z* 401.1967 (M + H)^+^, 423.1787 ( M + Na)^+^ (calculated for C_24_H_25_N_4_O_2_, 400.1899).

### 4.4. General Conditions for the Synthesis of Quinazolinone-3,6-(4H)-Diones Compound **4**, **5**, **6**, **7**, **8**, and **9**

In a closed vial anthranilic acid (**i**, 28 mg, 200 µmol), *N*-Boc-l-isoleucine (**x**, 44 mg, 200 µmol) for **4** and **5**, or *N*-Boc-l-methionine (**xi**, 46 mg, 200 µmol) for **6** and **7**, or *N*-Boc-*o*-Bn-Tyrosine (**xii**, 74 mg, 200 µmol) for **8** and **9**, and triphenylphosphite (63 µL, 220 µmol) were added along with 1 mL of dried pyridine. The vial was heated in heating block with stirring at 55 °C for 16–24 h. After cooling the mixture to room temperature, d-tryptophan methyl ester hydrochloride (**ii**) for **5**, and **7,**
L-tryptophan methyl ester hydrochloride (**vi**) for **4**, **6**, **8** and **9** (51 mg, 200 µmol) was added, and the mixture was irradiated in the microwave at the constant temperature at 220 °C for 1.5 min. Reaction mixtures were prepared in the same conditions and treated in parallel. After removing the solvent with toluene, the crude product was purified by flash column chromatography using hexane:EtOAc (60:40) as a mobile phase. The preparative TLC was performed using CH_2_Cl_2_:Me_2_CO (95:5) as mobile phase. The major compound appeared as a black spot with no fluorescence under the UV light (366 nm). The desirable compounds **4**, **5**, **6**, **7**, **8**, and **9** were collected as yellow solids. Before analysis, compounds were recrystallized from methanol. 

*(1R,4S)-4-((1H-indol-3-yl)methyl)-1-((S)-sec-butyl)-1,2-dihydro-6H-pyrazino[2,1-b]quinazoline-3,6(4H)-dione**(**4**).* Yield: 29.2 mg, 7.1%; *er* = 3:97; m.p: 220–221 °C; ${\left[\alpha \right]}_{D}^{30}$ = + 484.7 (*c* 0.037; CHCl_3_); *v_max_* (KBr) 3373, 3059, 2880, 1684, 1662, 1472, 1434, and 1261 cm^−1^; ^1^H NMR (300 MHz, CDCl_3_): *δ* 8. 38 (dd, 1H, *J* = 7.9 and 1.2 Hz, CH), 8.07 (br, 1H, NH-Trp), 7.78 (ddd, 1H, *J* = 8.4, 7.1, and 1.6 Hz, CH), 7.57 (d, *J* = 8.0 Hz, CH), 7.52 (d, 1H, *J* = 6.0 Hz, CH), 7.47 (d, 1H, *J* = 8.0 Hz, CH-Trp), 7.28 (d, 1H, *J* = 8.2 Hz, CH-Trp), 7.12 (t, 1H, *J* = 7.1 Hz, CH-Trp), 6.96 (t, 1H, *J* = 7.5 Hz, CH-Trp), 6.57 (d, 1H, *J* = 2.4 Hz, CH-Trp), 5.68 (dd, 1H, *J* = 5.2 and 2.7 Hz, CH*-Trp), 5.64 (s, 1H, NH-amide), 3.76 (dd, 1H, *J* = 14.8 and 2.7 Hz, CH_2_-Trp), 3.63 (dd, 1H, *J* = 14.9 and 5.3 Hz, CH_2_-Trp), 2.80 (d, 1H, *J* = 2.4 Hz, CH*-Ile), 2.36 (dt, 1H, *J* 14.9 and 7.5 Hz, CH*-Ile), 0.98 (m, 2H, CH_2_-Ile), 0.88 (d, 3H, *J* = 6.5 Hz, CH_3_-Ile), 0.64 (t, 3H *J* = 6.4 Hz, CH_3_-Ile); ^13^C NMR (75 MHz, CDCl_3_): *δ* 169.5 (C=O), 160.9 (C=O), 150.7 (C=N), 147.1 (C), 136.1 (C-Trp), 134.7 (CH), 127.2 (CH), 127.1 (CH), 127.0 (C-Trp), 126.9 (CH), 123.6 (CH-Trp), 122.8 (CH-Trp), 120.2 (C), 120.1 (C-Trp), 118.8 (CH-Trp), 111.1 (CH-Trp), 109.4 (C-Trp), 56.8 (CH*-Trp), 55.1 (CH*-Ile), 35.8 (CH*-Ile), 27.4 (CH_2_-Trp), 25.8 (CH_2_-Ile), 13.2 (CH_3_-Ile), 11.0 (CH_3_-Ile; (+)-HRMS-ESI *m/z* 401.1964 (M + H)^+^ (calculated for C_24_H_25_N_4_O_2_, 400.1899).

*(1S,4R)-4-((1H-indol-3-yl)methyl)-1-((S)-sec-butyl)-1,2-dihydro-6H-pyrazino[2,1-b]quinazoline-3,6(4H)-dione (**5**).* Yield: 27.2 mg, 6.6%; *er* = 3:97; m.p: 218–220 °C; ${\left[\alpha \right]}_{D}^{30}$ = −372.6 (*c* 0.034; CHCl_3_); *v_max_* (KBr) 3373, 3059, 2880, 1684, 1662, 1472, 1434, 1261 cm^−1^; ^1^H NMR (300 MHz, CDCl_3_): *δ* 8. 37 (dd, 1H, *J* = 7.9 and 1.2 Hz, CH), 8.07 (br, 1H, NH-Trp), 7.78 (ddd, 1H, *J* = 8.4, 7.1, and 1.6 Hz, CH), 7.57 (d, *J* = 8.0 Hz, CH), 7.52 (d, 1H, *J* = 6.0 Hz, CH), 7.47 (d, 1H, *J* = 8.0 Hz, CH-Trp), 7.28 (d, 1H, *J* = 8.2 Hz, CH-Trp), 7.12 (t, 1H, *J* = 7.1 Hz, CH-Trp), 6.96 (t, 1H, *J* = 7.5 Hz, CH-Trp), 6.57 (d, 1H, *J* = 2.4 Hz, CH-indole), 5.68 (dd, 1H, *J* = 5.2 and 2.7 Hz, CH*-Trp), 5.52 (s, 1H, NH-amide), 3.76 (dd, 1H, *J* = 14.8 and 2.7 Hz, CH_2_-Trp), 3.63 (dd, 1H, *J* = 14.9 and 5.3 Hz, CH_2_-Trp), 2.80 (d, 1H, *J* = 2.4 Hz, CH*-Ile), 2.37 (dt, 1H, *J* = 14.9 and 7.5 Hz, CH*-Ile), 0.88 (m, 2H, *J* = 6.7 Hz, CH_2_-Ile), 0.62 (d, 3H, *J* = 6.5 Hz, CH_3_-Ile), 0.46 (t, 3H *J* = 6.4 Hz, CH_3_-Ile); ^13^C NMR (75 MHz, CDCl_3_): *δ* 169.4 (C=O), 160.9 (C=O), 150.1 (C=N), 147.1 (C), 136.1 (C-Trp), 134.7 (CH), 127.2 (CH), 127.2 (CH), 127.0 (CH-Trp), 126.9 (CH), 123.6 (CH-Trp), 122.7 (CH-Trp), 120.2 (C), 120.1 (C-Trp), 118.8 (CH-Trp), 111.1 (CH-Trp), 109.4 (C-indol), 56.8 (CH*-Trp), 55.5 (CH*-Ile), 35.6 (CH*-Ile), 27.4 (CH_2_-Trp), 25.9 (CH_2_-Ile), 13.2 (CH_3_-Ile), 11.0 (CH_3_-Ile; (+)-HRMS-ESI *m/z* 401.1973 (M + H)^+^ (calculated for C_24_H_25_N_4_O_2_, 400.1899).

*(1R,4S)-4-((1H-indol-3-yl)methyl)-1-(2-(methylthio)ethyl)-1,2-dihydro-6H-pyrazino[2,1-b]quinazoline-3,6(4H)-diones* (**6**). Yield: 27 mg, 6.1%; m.p: 198–200.7 °C; ${\left[\alpha \right]}_{D}^{30}$ = +74.1 (*c* 0.045; CHCl_3_); *v_max_* (KBr) 3295, 3067, 2915, 1682, 1600, 1470, 770, and 697 cm^−1^; ^1^H NMR (300 MHz, CDCl_3_): *δ* 8. 37 (dd, 1H, *J* = 8.0 and 1.2 Hz, CH), 8.07 (br, 1H, NH-Trp), 7.78 (ddd, 1H, *J* = 8.4, 7.0, and 1.5 Hz, CH), 7.57 (d, 1H, *J = 2.5* Hz, CH), 7.53 (dd, 1H, *J* = 8.2 and 1.1 Hz, CH), 7.41 (d, 1H, *J* = 8.0 Hz, CH-Trp), 7.30 (d, 1H, *J* = 8.3 Hz, CH-Trp), 7.13 (t, 1H, *J* = 7.7 Hz, CH-Trp), 6.93 (t, 1H, *J* = 7.0 Hz, CH-Trp), 6.71 (d, 1H, *J* = 2.4 Hz, CH-Trp), 6.37 (s, 1H, NH-amide), 5.67 (dd, 1H, *J* = 5.2 and 3.0 Hz, CH*-Trp), 3.74 (dd, 1H, *J* = 15.0 and 3.0 Hz, CH_2_-Trp), 3.65 (dd, 1H, *J* = 14.9 and 5.3 Hz, CH_2_-Trp), 2.99 (dd, 1H, *J* = 6.6 and 3.6 Hz, CH*-Met), 2.49-2.13 (m, 4H, CH_2_-Met), 1.96 (s, 3H, CH_3_-Met); ^13^C NMR (75 MHz, CDCl_3_): *δ* 169.3 (C=O), 161.1 (C=O), 150.3 (C=N), 147.1 (C), 136.0 (C-Trp), 134.7 (CH), 127.2 (CH), 127.2 (C-Trp), 126.9 (CH), 123.5 (CH-Trp), 122.7 (CH-Trp), 120.2 (C), 120.2 (C-Trp), 118.7 (CH-Trp), 111.1 (CH-Trp), 109.5 (C-Trp), 57.2 (CH*Trp), 52.8 (CH*-Met), 30.7 (CH_2_-S-Met), 29.7 (CH_2_-Met), 27.2 (CH_2_-Trp), 15.3 (CH_3_-Met); HRMS-ESI m/z 419.1544 (M + H)^+^ (calculate for C_23_H_23_N_4_O_2_S, 418.1463).

*(1S,4R)-4-((1H-indol-3-yl)methyl)-1-(2-(methylthio)ethyl)-1,2-dihydro-6H-pyrazino[2,1-b]quinazoline-3,6(4H)-diones (**7**).* Yield: 34.8 mg, 7.9%; *er* = 49: 51; m.p.: 197–200 °C; ${\left[\alpha \right]}_{D}^{30}$ = −56.9 (*c* 0.041; CHCl_3_); *v_max_* (KBr) 3290, 3058, 2918, 2854, 1684, 1670, 1602, 773, and 695 cm^−1^; ^1^H NMR (300 MHz, CDCl_3_): *δ* 8. 37 (dd, 1H, *J* = 8.0 and 1.1 Hz, CH), 8.12 (br, 1H, NH-Trp), 7.78 (ddd, 1H, *J* = 8.4, 7.2, and 1.5 Hz, CH), 7.57 (d, *J* = 8.3 Hz, CH), 7.54 (d, 1H, *J* = 7.0 Hz, CH), 7.40 (d, 1H, *J* = 8.0 Hz, CH-Trp), 7.30 (d, 1H, *J* = 8.2 Hz, CH-Trp), 7.12 (t, 1H, *J* = 7.1 Hz, CH-Trp), 6.92(t, 1H, *J* = 7.0 Hz, CH-Trp), 6.71 (d, 1H, *J* = 2.4 Hz, CH-Trp), 5.67 (dd, 1H, *J* = 5.4 and 3.1 Hz, CH*-Trp), 6.54 (s, 1H, NH-amide), 3.73 (dd, 1H, *J* = 15.0 and 2.9 Hz, CH_2_-Trp), 3.65 (dd, 1H, *J* = 15.1 and 5.5 Hz, CH_2_-Trp), 3.01 (dd, 1H, *J* = 6.6 and 3.6 Hz, CH*-Met), 2.48-2.20 (m, 4H, CH_2_-Met), 1.96 (s, 3H, CH_3_-Met); ^13^C NMR (75 MHz, CDCl_3_): *δ* 169.5 (C=O), 161.0 (C=O), 150.6 (C=N), 146.9 (C), 136.1 (C-Trp), 134.7 (CH), 127.2 (C-Trp), 127.2 (CH), 126.9 (CH), 123.5 (CH-Trp), 122.6 (CH-Trp), 120.2 (C), 120.1 (C-Trp), 118.7 (CH-Trp), 111.2 (CH-Trp), 109.5 (C-Trp), 57.2 (CH*-Trp), 52.8 (CH*-Met), 30.7 (CH_2_-S-Met), 29.7 (CH_2_-Met), 27.2 (CH_2_-Trp), 15.3 (CH_3_-Met); (+)-HRMS-ESI *m/z* 419.1526 (M + H)^+^, (calculated for C_23_H_23_N_4_O_2_S, 418.1463).

*(1R,4S)-4-((1H-indol-3-yl)methyl)-1-(4-(benzyloxy)benzyl)-1,2-dihydro-6H-pyrazino[2,1-b]quinazoline-3,6(4H)-diones (**8***). Yield: 81.9 mg, 14.8%; *er* = 63:37; m.p.: 226.9–227.9 °C; ${\left[\alpha \right]}_{D}^{30}$ = +46.7 (*c* 0.05; CHCl_3_); *v_max_* (KBr) 3393, 3268, 2954, 1671, 1611, 1511, 1465, 1240, 772, and 697 cm^−1^; ^1^H NMR (300 MHz, CDCl_3_): *δ* 8. 39 (dd, 1H, *J* = 8.0 and 1.2 Hz, CH), 8.05 (br, 1H, NH-Trp), 7.80 (ddd, 1H, *J* = 8.5, 7.2 and 1.5 Hz, CH), 7.62 (d, *J* = 8.0 Hz, CH), 7.56 (dt, 1H, *J* = 7.3, 7.7, and 1.1 Hz, CH), 7.45-7.39 (m, 5H, CH-Bz), 7.32 (d, 2H, *J* = 8.0 Hz, CH-Trp), 7.17 (t, 1H, *J* = 7.5 Hz, CH-Trp), 6.88 (t, 1H, *J* = 7.5 Hz, CH-Trp), 6.76 (d, 2H, *J* = 9.0 Hz, CH-Tyr), 6.61 (d, 1H, *J* = 3.0 Hz, CH-Trp), 6.39 (d, 2H, *J* = 8.5 Hz, CH-Tyr), 5.64 (dd, 1H, *J* = 5.2 and 2.7, CH*-Trp), 5.35 (s, 1H, NH-amide), 5.05 (s, 2H, CH_2_-Bz), 3.76 (dd, 1H, *J* = 14.9 and 2.6 Hz, CH_2_-Trp), 3.67 (dd, 1H, *J* = 15.0 and 5.3 Hz, CH_2_-Trp), 3.52 (dd, 1H, *J* = 14.7 and 3.6 Hz, CH_2_-Tyr), 2.89 (dd, *J* = 11.1 and 3.6 Hz, CH*-Tyr), 2.46 (dd, 1H, *J* = 14.7 and 11.2 Hz, CH_2_-Tyr); ^13^C NMR (75 MHz, CDCl_3_): *δ* 169.2 (C=O), 160.7 (C=O), 157.9 (C-Tyr), 151.0 (C=N), 147.0 (C), 137.0 (C-Trp), 136.1(C-Bz), 134.8 (CH), 128.7 (CH-Tyr (2)), 128.6 (CH-Bz (2)), 128.0 (C-Tyr), 127.4 (CH-Bz), 127.3 (CH-Trp), 127.2 (CH), 127.1 (CH-Bz (2)), 127.0 (CH), 126.9 (CH), 123.8 (CH-Trp), 122.8 (CH-Trp), 120.6 (C), 120.4 (CH-Trp), 119.0 (CH-Trp), 115.5 (CH-Tyr (2H)), 111.1 (CH-Trp), 109.7 (C-Trp), 70.1 (CH_2_-Bz), 57.4 (CH*-Trp), 52.9 (CH*-Tyr), 37.1 (CH_2_-Tyr), 29.7 (CH_2_-Trp); (+)-HRMS-ESI *m/z* 541.2232 (M + H)^+^, (calculated for C_34_H_29_N_4_O_3_, 540.2161). 

*(1S,4S)-4-((1H-indol-3-yl)methyl)-1-(4-(benzyloxy)benzyl)-1,2-dihydro-6H-pyrazino[2,1-b]quinazoline-3,6(4H)-diones (**9***). Yield: 119.9 mg, 21.7%; *er* = 29:71; m.p.: 165.9–166.6 °C; ${\left[\alpha \right]}_{D}^{30}$ = +205.8 (*c* 0.076; CHCl_3_); ν_max_ (KBr) 3489, 3364, 2923, 1674, 1612, 1512, 1467, 1249, 774, and 695 cm^−1^; ^1^H NMR (300 MHz, CDCl_3_): *δ* 8. 42 (dd, 1H, *J* = 8.0 and 1.1 Hz, CH), 8.08 (br, 1H, NH-Trp), 7.83 (ddd, 1H, *J* 8.5, 7.1, and 1.5 Hz, CH), 7.66 (d, *J* 7.7 Hz, CH), 7.62-7.52 (m, 2H, CH), 7.39 (t, 4H, *J* = 2.6 Hz, CH-Bz), 7.35 (ddd, *J* = 6.2, 3.4, and 1.5 Hz, 1H, CH-Bz), 7.30 (d, 1H, *J* = 8.1 Hz, CH-Trp), 7.20 (td, 1H, *J* = 7.6 and 1.1 Hz, CH-Trp), 6.10 (td, 1H, *J* = 7.5 and 1.0 Hz, CH-Trp), 6.74 (d, 2H, *J* = 8.7 Hz, CH-Tyr), 6.60 (d, 1H, *J* = 2.3 Hz, CH-Trp), 6.23 (d, 2H, *J* = 8.6 Hz, CH-Tyr), 5.56 (t, 1H, *J* = 4.2 Hz, CH*-Trp), 5.55 (s, 1H, NH-amide), 4.99 (s, 2H, CH_2_-Bz), 4.33 (dt, 1H, *J* = 11.7 and 2.8 Hz, CH*-Tyr), 3.86 (dd, 1H, *J* =14.9 and 3.0 Hz, CH_2_-Trp), 3.80 (dd, 1H, *J* = 14.9 and 4.4 Hz, CH_2_-Trp), 2.95 (dd, 1H, *J* = 13.3 and 3.1 Hz, CH_2_-Tyr), 0.53 (dd, *J* = 13.1 and 11.9 Hz, CH_2_-Tyr); ^13^C NMR (75 MHz, CDCl_3_): *δ* 166.5 (C=O), 160.9 (C=O), 158.0 (C-Tyr), 150.2 (C=N), 147.2 (C), 136.9 (C-Trp), 135.8 (C-Bn), 134.9 (CH), 130.28 (CH-Tyr (2)), 128.6 (CH-Bz (2)), 128.1 (C-Tyr), 128.0 (CH-Bz), 127.7 (C-Trp), 127.4 (CH), 127.0 (CH-Bz (2)), 126.9 (CH), 123.5 (CH-Trp), 122.8 (CH-Trp), 120.5 (C), 120.2 (C-Trp), 119.5 (CH-Trp), 115.2 (CH-Tyr (2)), 111.4 (CH-Trp), 109.7 (C-Trp), 70.0 (CH_2_-Bz), 57.9 (CH*-Tyr), 56.8 (CH*-Trp), 42.0 (CH_2_-Tyr), 26.6 (CH_2_-Trp(+)-HRMS-ESI *m/z* 541.2221 (M + H)^+^, (calculated for C_34_H_29_N_4_O_3_, 540.2161).

### 4.5. General Conditions for the Synthesis of (1S)-4-((1H-indol-3-yl)methyl)-1-(4-hydroxybenzyl)-1,2-dihydro -6H-pyrazino[2,1-b]quinazoline-3,6(4H)-dione (**10** and **11**)

In an oven-dried round-bottomed flash equipped with Teflon-coated magnetic stir bar, a rubber septum, a glass stopper, and nitrogen gas inlet compound **8** or **9** (50 mg, 0.092 mmol) dissolved with anhydrous CH_2_Cl_2_ (5 mL) was added. After cooled the mixture to −78 °C, 1M boron trichloride in CH_2_Cl_2_ (190 µL, 0.19 mmol, 2eq) was added dropwise over 5 min at −78 °C. After stirring for 45 min at -78 °C, the mixture was quenched by syringe addition of CHCl_3_/MeOH (10/1, 10 mL) at -78 °C and was warmed to ambient temperature. The solvent was evaporated, and the content was purified by flash column chromatography using hexane: EtOAc (6:4) as mobile phase. Compounds **10** or **11** were obtained as pale yellow solids. Before analysis, compounds were recrystallized from methanol. 

*(1R,4S)-4-((1H-indol-3-yl)methyl)-1-(4-hydroxybenzyl)-1,2-dihydro-6H-pyrazino[2,1-b]quinazoline-3,6(4H)-dione (**10***). Yield: 12.5 mg, 30%; *er* = 36:65; m.p.: 134–136 °C; ${\left[\alpha \right]}_{D}^{30}$ = +39.5 (*c* 0.034; CH_3_OH); *v_max_* (KBr) 3427, 1677, 1603, 1515, 1468, 1159, 1025, 998, and 765 cm^−1^; ^1^H NMR (300 MHz, DMSO-d_6_): *δ* 10.48 (s, 1H, NH-Trp), 8.89 (d, 1H, *J* = 4.9 Hz, OH-Tyr), 8.32 (dd, 1H, *J* = 8.0 and 1.2 Hz, CH), 7.81 (dd, 1H, *J* = 7.7 and 1.6 Hz, CH), 7.61 (d, *J* = 7.9 Hz, CH), 7.56 (dd, 1H, *J* = 12.7 and 7.2 Hz, CH), 7.39 (dd, 1H, *J* = 8.0 and 5.4 Hz, CH-Trp), 7.34 (dt, 1H, *J* = 8.1 Hz, CH), 7.13 (dd, 1H, *J* = 13.4 and 6.7 Hz, CH-Trp), 6.84 (dd, 1H, *J* = 13.4 and 6.7 Hz, CH-Trp), 6.60 (d, 1H, *J* = 3.4 Hz, CH-Trp), 6.58(dd, 2H, *J* = 8.5 and 6.0 Hz, CH-Tyr), 6.53 (d, 1H, *J* = 4.4 Hz, NH-amide), 6.46 (dd, 2H, *J* = 7.9 and 5.7 Hz, CH-Tyr), 5.44 (dd, 1H, *J* = 5.0 and 2.8 Hz, CH*-Trp), 3.64 (dd, 1H, *J* = 14.8 and 2.7 Hz, CH_2_-Trp), 3.56 (dd, 1H, *J* = 14.9 and 5.3 Hz), 3.26 (dt, *J* = 13.4 and 3.6 Hz, CH_2_-Tyr), 3.03 (dt, 1H, *J* = 8.9 and 4.7 Hz, CH*-Tyr), 2.67 (dt, 1H, *J* = 13.3 and 10.5 Hz); ^13^C NMR (75 MHz, CDCl_3_): *δ* 168.1 (C=O), 160.0 (C=O), 150.5 (C-Tyr), 146.3 (C), 135.8 (C-Trp), 134.1 (CH), 129.4 (CH-Tyr (2)), 126.7 (C-Trp), 126.6 (CH), 126.4 (CH), 126.1 (CH), 124.9 (C-Tyr), 123.7 (CH-Trp), 121.2 (CH-Trp), 119.6 (CH-Trp), 118.9 (CH-Trp), 115.0 (C-Tyr), 111.2 (CH-Trp), 108.0 (C-Trp), 56.6 (CH*-Trp), 52.8 (CH*-Tyr), 36.3 (CH_2_-Tyr), 26.5 (CH_2_-Trp); (+)-HRMS-ESI *m/z* 451.1766 (M + H)^+^, 473.1576 (M + Na)^+^ (calculated for C_27_H_23_N_4_O_3_, 450.1692).

*(1S,4S)-4-((1H-indol-3-yl)methyl)-1-(4-hydroxybenzyl)-1,2-dihydro-6H-pyrazino[2,1-b]quinazoline-3,6(4H)-dione**(**11***). Yield: 25.7 mg, 68.7%; *er* = 61:39; m.p.: 102–103 °C; ${\left[\alpha \right]}_{D}^{30}$ = +75.9 (*c* 0.079; CH_3_OH); *v_max_* (KBr) 3428, 2927, 1667, 1610, 1592, 1474, 1337, 1232, 772, and 699 cm^–1^; ^1^H NMR (300 MHz, DMSO-d_6_): *δ* 10.54 (s, 1H, NH-Trp), 8.86 (s, 1H, OH-Tyr), 8.34 (dd, 1H, *J* = 8.0 and 1.2 Hz, CH), 7.83 (dd, 1H, *J* = 7.7, and 1.6Hz, CH), 7.62 (d, *J* = 7.9 Hz, CH), 7.56 (dt, 1H, *J* = 7.6, 7.6, and 0.6 Hz, CH), 7.49 (d, 1H, *J* = 7.9 Hz, CH-Trp), 7.30 (d, 1H, *J* = 8.1 Hz, CH), 7.20 (d, 1H, *J* = 3.3 Hz, NH-amide), 7.09 (dt, 1H, *J* = 7.6 and 0.6 Hz, CH-Trp), 6.94 (dt, 1H, *J* = 7.5 and 0.5 Hz, CH-Trp), 6.66 (d, 1H, *J* = 2.3 Hz, CH-Trp) 6.57(d, 2H, *J* = 5.6 Hz, CH-Tyr), 6.45 (d, 2H, *J* = 8.4 Hz, CH-Tyr), 5.37 (dd, 1H, *J* = 5.2 and 3.3 Hz, CH*-Trp), 4.36 (dt, 1H, *J* = 10.5 and 3.4 Hz, CH*-Tyr), 3.63 (dd, 1H, *J* = 14.8 and 3.1 Hz, CH_2_-Trp), 3.55 (dd, 1H, *J* = 14.9 and 5.4 Hz, CH_2_-Trp), 2.69 (dd, *J* = 13.4 and 3.6 Hz CH_2_-Tyr), 0.86 (dd, 1H, *J* = 13.3 and 10.5 Hz, CH_2_-Tyr); ^13^C NMR (75 MHz, CDCl_3_): *δ* 165.8 (C=O), 160.2 (C=O), 155.8 (C-Tyr), 150.4 (C=N), 146.7 (C), 135.6 (C-Trp), 134.2 (CH), 129.9 (CH-Tyr (2)), 127.4 (CH-Trp), 126.3 (CH), 126.2 (CH), 126.1 (CH-Trp), 125.9 (C-Tyr), 123.7 (CH-Trp), 121.1 (CH-Trp), 119.5 (CH-Trp), 118.7 (CH-Trp), 114.9 (CH-Tyr (2)), 111.2 (CH-Trp), 108.0 (C-Trp), 57.2 (CH*-Trp), 56.3 (CH*-Tyr), 41.8 (CH_2_-Tyr), 26.1 (CH_2_-Trp); (+)-HRMS-ESI *m/z* 451.1771 (M + H)^+^, 473.1564 (M + Na)^+^ (calculated for C_27_H_23_N_4_O_3_, 450.1692).

### 4.6. General Conditions for the Synthesis of (1S,4R)-4-((1H-indol-3-yl)methyl)-1-(4-(benzyloxy)benzyl)-8,10 -dichloro-1,2-dihydro-6H-pyrazino[2,1-b]quinazoline-3,6(4H)-dione (**12**)

In a closed vial 3,5-dichloro anthranilic acid (**xiii**, 41 mg, 200 µmol), *N*-Boc-*ο*-Bn-Tyrosine (**xii**, 74 mg, 200 µmol), and triphenylphosphite (63 µL, 220 µmol) were added along with 1 mL of dried pyridine. The vial was heated in heating block with stirring at 55 °C for 16–24 h. After cooling the mixture to room temperature, d-tryptophan methyl ester hydrochloride (**ii**, 51 mg, 200 µmol) was added, and the mixture was irradiated in the microwave at the constant temperature at 220 °C for 1.5 min. Reaction mixtures were prepared in the same conditions and treated in parallel. After removing the solvent with toluene, the crude product was purified by flash column chromatography using hexane: EtOAc (60:40) as a mobile phase. The preparative TLC was performed using CH_2_Cl_2_:Me_2_CO (95:5) as mobile phase. The major compound appeared as a black spot with no fluorescence under the UV light (366 nm). Compound **12** was collected as orange solids. Before analysis, compound was recrystallized from methanol. Yield: 26.2 mg, 2.2%; *er* 67:33; m.p.: 233–235 °C; ${\left[\alpha \right]}_{D}^{30}$ = +244.44 (*c* 0.045; CH_3_OH); ν_max_ (KBr) 3424, 3334, 2921, 1681, 1593, 1511, 1455, 1247,1012, 695, and 420 cm^−1^; ^1^H NMR (300 MHz, CDCl_3_): *δ* 8.26 (d, 1H, *J* = 2.4 Hz, CH), 8.09 (br, 1H, NH-Trp), 7.85 (d, 1H, *J* = 2.4 Hz, CH), 7.44 (dd, 5H, *J* = 6.4 and 2.0 Hz, CH-Bz), 7.39 (d, 2H, *J* = 8.2 Hz, CH-Trp), 7.19 (t, 1H, *J* = 7.7 Hz, CH-Trp), 6.96 (t, 1H, *J* = 7.5 Hz, CH-Trp), 6.75 (d, 2H, *J* = 8.7 Hz, CH-Tyr), 6.62 (d, 1H, *J* = 2.3 Hz, CH-Trp), 6.40 (d, 2H, *J* = 8.5 Hz, CH-Tyr), 5.56 (dd, 1H, *J* = 5.2 and 2.6, CH*-Trp), 5.41 (s, 1H, NH-amide), 5.06 (s, 2H, CH_2_-Bz), 3.77 (dd, 1H, *J* = 15.0 and 2.6 Hz, CH_2_-Trp), 3.62 (dd, 1H, *J* = 15.6 and 4.1 Hz, CH_2_-Trp), 3.50 (dd, 1H, *J* = 17.5 and 4.4 Hz, CH_2_-Tyr), 2.95 (dd, *J* = 10.8 and 3.6 Hz, CH*-Tyr), 2.51 (dd, 1H, *J* = 14.8 and 10.9 Hz, CH_2_-Tyr); ^13^C NMR (75 MHz, CDCl_3_): *δ* 168.7 (C=O), 159.2 (C=O), 158.0 (C-Tyr), 152.0 (C=N), 142.4 (C), 136.9 (C-Bz), 136.1(C-Trp), 135.1 (CH), 133.2 (C-Cl), 132.6 (C-Cl), 129.6 (CH-Tyr (2)), 128.7 (CH-Bz (2)), 128.1 (CH-Bz), 127.4 (CH-Bz(2)), 126.9 (C-Trp), 126.6 (C-Tyr), 125.2 (CH), 123.7 (CH-Trp), 122.9 (CH-Trp), 122.4 (C), 118.9 (CH-Trp), 115.5 (CH-Tyr (2H)), 111.2 (CH-Trp), 109.4 (C-Trp), 70.1 (CH_2_-Bz), 57.6 (CH*-Trp), 53.1 (CH*-Tyr), 36.9 (CH_2_-Tyr), 27.1 (CH_2_-Trp). (+)-HRMS-ESI *m/z* 609.1427 (M + H)^+^, (calculated for C_34_H_27_N_4_O_3_Cl_2_, 608.1382). 

### 4.7. Neuroprotection Assay

Compounds **1**, **2**, **4**, **5**, **6**, **7**, **8**, **9**, **10, 11,** and **12c** together with fiscalin B (**3**) were assayed in co-treatment with rotenone at eight concentrations per triplicate. The SH-SY5y cells (ATCC Ref.: CRL-2266) were seeded at a density of 40,000/well in a 96-well plate and were incubated in a humidified atmosphere at 37 °C with 5% CO_2_ for overnight. The compounds were dissolved in DMSO at the concentration of 10 mM and the higher concentration assayed was 100 µM. The negative control was 1% DMSO. After 24 h of co-treatment, plates were treated with MTT (3-(4,5-dimethylthiazol-2-yl)-2,5-diphenyltetrazolium bromide) at 5 µg/mL in Minimum Essential Medium Eagle (MEM) for 3 h at the standard culture condition. Then, DMSO was added to the plates to solubilizing the formazan crystal formed in viable cells and plates were put in the stirring for 5 min to homogenize the solution. Absorbance at 570 nm was measured by VICTOR Multilabel Plate Reader (PerkinElmer).

### 4.8. Screening Test for Antitumor Activity

Compounds **1**, **2**, **4**, **5**, **6, 7, 8**, **10**, and **11** were reconstituted in sterile DMSO to the final concentration of 60 mM, and several aliquots were made and stored at −20 °C to avoid repeated freeze-thaw cycles. For experiments, the compounds were freshly diluted in medium to the desired concentration. Screening for tumor cell growth inhibitory activity was carried out in three human tumor cell lines (NCI-H460, BxPC3 and PANC1), with the sulforhodamine B (SRB) assay, as previously described [32]. Briefly, tumor cells were plated in 96-well plates, incubated at 37 °C for 24 h, and then treated for 48 h with 5 serial dilutions (1:2) of each compound (ranging from 25 μM to 1.5625 μM, 150 μM to 9.375 μM or 200 μM to 12.5 μM, depending on the compound and for reasons related with solubility). The effect of the vehicle solvent (DMSO) was also analyzed as a control. Cells were fixed with 10% ice-cold trichloroacetic acid, washed with water, and stained with SRB. Finally, the plates were washed with 1% acetic acid and the bound SRB was solubilized with 10 mM Tris Base. Absorbance was measured in a microplate reader (Synergy Mx, Biotek Instruments Inc., Winooski, VT, USA) at 510 nm. For each compound, the corresponding GI_50_ (concentration which inhibited 50% of net cell growth) was determined, as previously described [33]. 

### 4.9. Testing Effect of Compounds on Non-malignant Breast Cells

Compounds **5**, **7, 10** and **11** (which presented simultaneously the best neuroprotection and antitumor effects) were tested against the non-malignant MCF-12A human breast epithelial cells. For that, cells were incubated with specific concentrations of each compound (corresponding to approximately the highest GI_50_ concentration obtained in the antitumor activity screening) for 48 h, followed by removal of the compound, addition of new medium to the cells and then 5 more days in culture. At the end of the 7 days in total, the sulforhodamine B (SRB) assay was performed, as previously described [32].

## 5. Conclusions

New quinazolinone alkaloid derivatives with *anti* and *syn* stereochemistry were synthetized by combining both a one-pot microwave-assisted reaction and a multi-step approach. Interestingly, fumiquinazoline G (**1**) presented a better antitumor activity in all the tumor cell lines tested, with GI_50_ values lower than 20 µM. The antitumor activity of the remaining compounds was not relevant, with GI_50_ values higher than 20 µM in the tested cell lines. The effect of the synthesized compounds in the growth of the tested non-malignant cells was smaller than the effect on the studied tumor cells. It is worth noting that among the compounds tested, only **1**, **3**, **5**, and **7** showed potential for neuroprotection in a PD in vitro model. This finding highlights new insights into marine natural products belonging to the proteomimetic quinazolinone alkaloids.

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
