# Peer review of "Synthesis of New Proteomimetic Quinazolinone Alkaloids and Evaluation of Their Neuroprotective and Antitumor Effects"

_molecules, 2019, doi:10.3390/molecules24030534_

Reviewer 1 Report

Prof. Sousa and co-workers in this manuscript reported synthesis of quinazolinone alkaloids and derivatives, as well as evaluation of their biological activities regarding neuroprotection and anti-proliferation of cancer cells. Quinazolinone alkaloid is a class of fused heterocyclic natural products which are known for their diversified biological activities. Therefore, they attract the attention from synthetic and medicinal chemists for decades. Numerous reports regarding total synthesis of this family members were published. Studies are still undergoing to not only simplify and optimize synthetic process, but also identify their primary cellular targets. Prof. Sousa's research team used general synthetic scheme towards 11 analogues, the structures of which were fully characterized. The neuroprotective activity was evaluated by using toxin-treated cells. The anti-proliferative effect was rated in not only tumor cell lines, but also non-malignant cells. Based on the results, SAR trend was concluded, which can be used to guide further optimization. This reviewer thus recommends acceptance after minor revision:

1. The authors provided adequate evidence to characterize the stereochemistry of compounds. Though HMBC showed different coupling pattern between diastereomers, some 2D-NOE studies may still needed as confirmative information. It would be excellent to see some X-ray crystal structures, but not required.

Author Response

We understand the reviewer’s concern. For fiscalin B (3) we have performed NOESY experiments and added these data to the revised version of this manuscript (Please see supplementary material). We have also added the sentence:  “The NOESY spectrum of fiscalin B (3) exhibited correlations from the protons of the methyl groups on C-1' to H-4, as well as from H-1 to H-2 of the amide group [17]”, thus definitely “confirming the anti configuration”.

Reviewer 2 Report

Just a few comments below will be improved the article:

Page 1, Line 28, compound should be compounds.

Page 6, Line 171, compound should be compounds

Author Response

Done.

Reviewer 3 Report

The authors should further clarify their stereochemical considerations for the molecules synthesised. Starting with enantiopure amino acids (as inferred in Scheme 1), it is expected that stereochemistry would be retained during synthesis which should lead to enantiopure substrates, however for most cases, a very moderate enantiomeric ratio is obtained. If one or both of the amino acids are epimerising or racemising during the synthetic process, then this would in turn lead to four possible compounds (two diastereomeric pairs and four possible enantiomers). There is no indication that diastereomeric pairs were observed or considered for this process. 

Also in regards to the oxazole intermediate proposed in Scheme 1, Method A, mechanistically it is unlikely that the imine as drawn would form. More likely, the tryptophan nitrogen undergoes a ring-closing condensation process with the proximal amide to form the central ring.

Author Response

The authors should further clarify their stereochemical considerations for the molecules synthesised. Starting with enantiopure amino acids (as inferred in Scheme 1), it is expected that stereochemistry would be retained during synthesis which should lead to enantiopure substrates, however for most cases, a very moderate enantiomeric ratio is obtained. If one or both of the amino acids are epimerising or racemising during the synthetic process, then this would in turn lead to four possible compounds (two diastereomeric pairs and four possible enantiomers). There is no indication that diastereomeric pairs were observed or considered for this process. 

Reply: Our results are in accordance with the previously described results obtain by Liu et al. (reference 19) in which an investigation of the impact of the reaction conditions on epimerization resulted in similar outcomes, i.e., a partial epimerization was observed at both C-1 and C-4 and it is higher with bulky substituents at C-1 (which is the case of the substituents presented herein).

Also in regards to the oxazole intermediate proposed in Scheme 1, Method A, mechanistically it is unlikely that the imine as drawn would form. More likely, the tryptophan nitrogen undergoes a ring-closing condensation process with the proximal amide to form the central ring.

Reply: We understand the reviewer’s concern and we therefore removed the intermediate from Scheme 1. Nevertheless, there are reports describing the dehydration of β-keto amide to oxazole, and by Wipf’s conditions the desired product is formed [1]. Later, the oxazole was identified as an oxazine [2]. The mechanism for the conversion of the oxazine to quinazoline upon piperidine treatment is also described [3].  

1.            Wang, H.; Ganesan, A. Total synthesis of the quinazoline alkaloids (-)-fumiquinazoline g and (-)-fiscalin b. Journal of Organic Chemistry 1998, 63, 2432-2433.

2.            Wang, H.; Ganesan, A. Total synthesis of the fumiquinazoline alkaloids: Solution-phase studies. Journal of Organic Chemistry 2000, 65, 1022-1030.

3.            Snider, B.B.; Zeng, H. Total synthesis of (-)-fumiquinazolines a, b, c, e, h, and i. Approaches to the synthesis of fiscalin a. Journal of Organic Chemistry 2003, 68, 545-563.